# Clonal Dissemination of Pandrug-Resistant *Klebsiella pneumoniae* ST392KL27 in a Tertiary Care Hospital in Mexico

**DOI:** 10.3390/ijms26168047

**Published:** 2025-08-20

**Authors:** Iliana Alejandra Cortés-Ortíz, Enoc Mariano Cortés-Malagón, Eduardo García-Moncada, Gustavo Acosta-Altamirano, Jesús Alejandro Pineda-Migranas, Karen Lizzet García-Prudencio, Edgar Mendieta-Condado, Adnan Araiza-Rodríguez, Alejandra Yareth Bonilla-Cortés, Mónica Sierra-Martínez, Juan Carlos Bravata-Alcántara

**Affiliations:** 1Genetics and Molecular Diagnostics Laboratory, Hospital Juárez de México, Mexico City 0776, Mexico; iliancortes@yahoo.com.mx (I.A.C.-O.); eduardo.garcia.moncada@gmail.com (E.G.-M.); 2Research Division, Hospital Juárez de México, Mexico City 07760, Mexico; emcortes@cinvestav.mx; 3Hospital General de México, Mexico City 06726, Mexico; mq9903@live.com.mx; 4Facultad de Estudios Superiores Cuautitlán, Universidad Nacional Autónoma de México, Cuautitlán Izcalli 54740, Mexicokl.gprudencio@gmail.com (K.L.G.-P.); 5State Public Health Laboratory, Jalisco Health Services, Zapopan 45177, Mexico; emendiet76@gmail.com; 6Unit of Technological Development and Molecular Research, Instituto de Diagnóstico y Referencia Epidemiológicos (InDRE) “Dr. Manuel Martínez Báez”, Secretaría de Salud, Mexico City 01480, Mexico; otto_twister84@yahoo.com.mx; 7School of Medicine and Homeopathy, Instituto Politécnico Nacional, Mexico City 07320, Mexico; yarethcortes1998@gmail.com; 8Health Research Unit, Hospital de Alta Especialidad de Ixtapaluca, Servicios de Salud del Instituto Mexicano del Seguro Social para el Bienestar (IMSS-BIENESTAR), Ixtapaluca 56530, Mexico

**Keywords:** *Klebsiella pneumoniae*, ST392, pandrug-resistance, hospital outbreak, whole-genome sequencing, CRISPR-Cas system, integrons, mobile genetic elements

## Abstract

The global emergence of multidrug- and pandrug-resistant *Klebsiella pneumoniae* poses a critical threat to public health, particularly in hospital settings. This study describes a nosocomial outbreak caused by *K. pneumoniae* in a tertiary-care hospital in Mexico and provides a comprehensive genomic analysis of six clinical isolates. All isolates exhibited pandrug resistance, including carbapenems and colistin. Whole-genome sequencing identified 37 antimicrobial resistance genes, including *bla*NDM-1, *bla*OXA-1, *bla*CTX-M-15, and a *pmr*B R256G mutation associated with colistin resistance. Two conjugative plasmids (pAA046 and pAA276) carried multiple resistance genes and mobile genetic elements. Although all isolates harbored CRISPR-Cas type I-E systems, no spacers matched resistance plasmids, suggesting functional inactivity. Capsular typing identified the KL27 locus with the *wzi*187 allele. Phylogenetic and cgMLST analyses confirmed clonal dissemination and close genetic relatedness to strains from Europe and the USA. Despite the absence of classical hypervirulence markers, the presence of *kfu*, *fimH*, and *mrkD* genes indicates adaptation to the hospital environment. These findings confirm the clonal spread of pandrug-resistant *K. pneumoniae* ST392-KL27 in a Mexican hospital, underscoring the role of plasmid-mediated resistance and the potential for global dissemination.

## 1. Introduction

Antimicrobial resistance is one of the most pressing global public health threats, particularly in hospital environments where multidrug-resistant (MDR) pathogens compromise the effective treatment of severe infections [1]. *Klebsiella pneumoniae* has emerged as a significant cause of nosocomial infections, including ventilator-associated pneumonia, sepsis, complicated urinary tract infections, and surgical site infections [2]. The rise of MDR and, in some cases, pandrug-resistant (PDR) strains—defined as isolates that are non-susceptible to all major classes of antibiotics, including colistin—has intensified this clinical and epidemiological crisis [3].

In Mexico, the situation is particularly alarming. *K. pneumoniae* is among the most frequently isolated pathogens in bloodstream infections and shows increasing resistance to third- and fourth-generation cephalosporins, fluoroquinolones, and carbapenems [4]. Recent reports have described hospital outbreaks caused by carbapenemase-producing strains, especially those carrying the *bla*NDM gene, which are associated with high mortality rates and limited therapeutic options [5].

Globally, high-risk clones of carbapenem-resistant *K. pneumoniae* include ST11, ST14, ST17, ST147, ST512, ST258, and ST230, which are primarily grouped into clonal groups, such as CG14, CG17, CG258, and CG147, which are widely disseminated in healthcare environments [6]. In Latin America, however, *bla*NDM harboring strains remain relatively uncommon and poorly characterized, underscoring the need for enhanced genomic surveillance [7].

Next-generation sequencing (NGS) has become a crucial tool for the genomic surveillance of resistant pathogens, enabling the precise detection of resistance genes, mobile genetic elements, and high-resolution phylogenetic inference [8]. CRISPR-Cas systems, which function as adaptive immune mechanisms against foreign DNA, have also emerged as valuable tools in tracing horizontal gene transfer events and clonality within *K. pneumoniae* outbreaks [9]. The analysis of CRISPR spacer content has been proposed as a method to monitor plasmid transmission and to predict trends in resistance acquisition [10].

Although the isolates examined in this study were collected in 2016, their genomic characterization remains highly relevant in 2025. Carbapenemase-producing *K. pneumoniae* strains, particularly those harboring the *bla*NDM gene, continue to pose a serious threat in healthcare settings throughout Latin America. Recent genomic surveillance reports have been published from Peru, Chile, Colombia, and Jamaica [11,12,13,14,15]. However, comprehensive genomic analyses addressing resistance mechanisms, mobile genetic elements, virulence traits, and CRISPR-Cas systems remain scarce. This retrospective study provides critical baseline data to understand the persistence and evolution of high-risk clones over time. These insights remain novel and relevant for genomic epidemiology and public health efforts in the region.

This study aims to perform a comprehensive genomic characterization of PDR *K. pneumoniae* strains isolated from a tertiary care hospital in Mexico, focusing on resistance gene profiles, CRISPR-Cas system diversity, and phylogenetic relationships to elucidate the clonal dissemination patterns within the hospital environment.

## 2. Results

### 2.1. Clinical Characteristics of Patients and Microbiological Identification of Isolates

For over four months, six patients were identified with pandrug-resistant *K. pneumoniae*. The outbreak began with the first isolate, strain HJM-EMC466, which was detected on 21 June 2016. After that, three strains were recovered from patients in the intensive care unit (ICU): strain HJM-NEE392 on 8 July, strain HJM-ERG332 on 2 August, and strain HJM-ICO381 on 22 August. Following that, strain HJM-CBA208 was isolated in the internal medicine department on 11 September, and strain HJM-CCQ432 was identified in the orthopedics department on 14 September. Overall, these cases accounted for 0.137% of the total 4355 Gram-negative bacterial cultures processed in 2016. The infections included bloodstream infections (*n* = 4), ventilator-associated pneumonia (*n* = 1), and multiple liver abscesses (*n* = 1). The patients were aged 28, 32, 34, 45, 53, and 81 years, and all had received prior antibiotic treatment with carbapenems or third-generation cephalosporins (Table 1).

The outbreak was initially detected through routine surveillance by the hospital’s epidemiology service, which identified a sudden increase in cases caused by pandrug-resistant *K. pneumoniae* across the ICU and orthopedics units. The ICU has a capacity of nine beds, with a nurse-to-patient ratio of 1:1 and a physician-to-patient ratio of 1:2. By contrast, both internal medicine and orthopedics have higher patient loads, with six patients per nurse and nine patients per physician. Patient flow was carefully managed to contain the spread of infection, implementing strict measures including isolation of the infected patients, adherence to hand hygiene protocols, and regular disinfection cycles. These infection control efforts were effective, as evidenced by the subsequent decrease in new cases following intervention.

The outbreak was associated with high mortality, as five of the six patients died. A retrospective review of medical records identified the index case as a patient presenting with liver abscess and pneumonia. Subsequent cases developed severe invasive infections. All isolates were identified as *K. pneumoniae* via the PHOENIX-100 automated system with high confidence scores (>99.9%).

### 2.2. Antimicrobial Resistance Profile

All the isolates exhibited a PDR phenotype. Susceptibility testing identified resistance to carbapenems (imipenem, meropenem, and ertapenem), third- and fourth-generation cephalosporins (ceftazidime and cefepime), aminoglycosides (amikacin, gentamicin, and tobramycin), and colistin (100%).

Phenotypic confirmation of carbapenemase production was performed via the TEST/CARBA-5 assay and validated through whole-genome sequencing. The following resistance genes were identified: carbapenemases, *bla*NDM-1 (100%) and *bla*OXA-1 (100%), and extended-spectrum β-lactamases (ESBLs), *bla*TEM-1B (100%) and *bla*CTX-M-15 (83%).

The following genes conferring resistance to aminoglycosides were also detected: *rmtC*, *aadA5*, *aph(6′)-Id*, *aph(3′)-Ib* (100%), and *aac(6′)-Ib-cr* (83%). The fluoroquinolone resistance genes included *oqx*A, *oqx*B, and *qnr*B1 (100%), along with *aac(6′)-Ib-cr* (83%). Other resistance genes identified included *mph*A (83%) and *erm*B (66%) for macrolides, *cat*B3 (100%) for phenicols, *sul*1 and *sul*2 (100%) for sulfonamides, *tet*A (100%) for tetracyclines, *fos*A (100%) for fosfomycin, and *dfr*A17 and *dfr*A14 (100%) for trimethoprim. Although no *mcr* genes were detected, a Kleborate analysis identified the *pmrB* R256G mutation, which is known to confer colistin resistance.

### 2.3. Genomic Collection and Clonal Analysis

Whole genome sequencing data were deposited in GenBank (NCBI) under BioProject ID PRJNA764253. The assembly quality data are listed in Appendix A.

All the isolates belonged to type ST392 (allelic profile 3-4-6-1-7-4-40), corresponding to the CC147 clonal complex. An extended MLST analysis identified the subgenotype scgST-64, except for HJM-ICO381, which also carried the scgST51652 subgenotype.

Capsular typing via the BIGSdb-Kp platform assigned all the isolates to KL27, with the *wzi*187 allele and O-locus O4, which encode the polysaccharide capsule and the lipopolysaccharide O antigen, respectively. These findings were consistent across all the isolates.

### 2.4. Resistome and Mobile Genetic Elements

A total of 37 distinct antibiotic resistance genes were identified across the six *K. pneumoniae* via AMRFinderPlus, covering resistance to β-lactams, aminoglycosides, fluoroquinolones, macrolides, sulfonamides, tetracyclines, trimethoprim, phenicols, and rifampin (Figure 1).

All isolates shared a core resistome that included *bla*NDM-1, *bla*SHV-11, and *bla*TEM-1, which are key determinants of carbapenem and extended-spectrum β-lactam resistance. These genes are found on both chromosomes and plasmids, underscoring the genomic plasticity and dissemination potential of resistance elements.

Plasmid-borne genes were predominant for β-lactamases and aminoglycoside-modifying enzymes, whereas chromosomal localization was more frequent for *oqx*AB, fosA, and mutations in *gyr*A and *par*C. A match strength analysis identified high-confidence hits for most genes, although a few, such as *mph*A and *val*S, presented relatively low identity, suggesting possible sequence variation.

IntegronFinder detected CALIN-type integrons (cluster of *att*C sites lacking an integrase) in the strains HJM-EMC466, HJM-NEE391, HJM-ICO381, and HJM-CBA208. These structures carried *aad*A5 and *dfr*A17, but lacked the *int*l1 gene, suggesting a potential for mobilization. A BLAST (BLAST+ 2.17.0) analysis identified 100% identity with class 1 integron-bearing plasmid MZ670313.1. This high degree of sequence conservation suggests that the integrons in our isolates likely share a common evolutionary origin with those carried by MZ670313.1. Furthermore, this finding implies potential horizontal gene transfer events and the circulation of conserved integron-bearing plasmids within hospital environments, contributing to the dissemination of antimicrobial resistance determinants. Notably, In0-type integrons were found in the strains HJM-ERG332 and HJM-CCQ423, suggesting past cassette acquisition events. Notably, *bla*NDM-1 and *bla*OXA-1 were not associated with integrons, implying that other mobile elements carry them.

### 2.5. Genomic Resistance and Mobile Genetic Elements

Between three and five plasmids were identified in the analyzed isolates via AMRFinderPlus and MOBsuite (Table 2). Two large conjugative plasmids—pAA046 (~96 kb) and pAA276 (~172 kb)—were consistently present and harbored most AMR genes.

The plasmid pAA046 carried *bla*NDM-1, *rmt*C, and *sul*1 (Figure 2) in selected strains. The plasmid pAA276 carried a broader set of AMR genes, including *aac(6′)-Ib-cr5*, *aph(3″)-Ib*, *aph(6)-Id*, *bla*CTX-M15, *bla*OXA-1, *bla*TEM-1, *qnr*B1, *sul*2 and *tet*A (Figure 3); *cat*B3 was also present in several isolates.

Both plasmids were conjugative, harboring MOBF relaxases and IncFIB and IncFII replicons, indicating a broad host range and horizontal transfer capacity. A MASH distance analysis identified substantial similarity to reference plasmids CP010390 (distance ~0.0024 with pAA276) and CP022350 (distance ~0.00225–0.0044 with pAA046).

In addition to conjugative plasmids, small non-mobilizable plasmids have also been identified. A frequently observed example was the ~4 kb ColRNAI-type plasmid (cluster AA103), which lacked detectable relaxase genes and was classified as non-mobilizable, with extremely low MASH distances (~0.0002) to CP024435. Other non-mobilizable elements, such as B749, ADO94, and AC249, presented larger MASH distances (up to ~0.058), reflecting greater sequence divergence and less structural conservation than the core conjugative elements.

Plasmid pAA276 harbored 90 mobile genetic elements (MGEs), including 32 integration/excision, 15 replication/recombination/repair, 7 stability/defense, and 36 associated with transfer functions. These features are consistent with the presence of a class 1 integron within this plasmid. The pAA046 and pAA276 plasmids are both classified as conjugative, underscoring their role in the horizontal dissemination of multidrug resistance.

In strain HJM-EMC466, plasmid pAA046 also carries genes spanning multiple functional categories, including 12 related to integration/excision, 11 related to replication, recombination, and repair, 2 related to phage functions, 9 related to stability and defense, and 29 related to transfer. The distribution of these genes varied among the isolates. Mapping against the CARD database identified that the *bla*NDM-1 gene, detected in all the isolates, was flanked by genes involved in plasmid maintenance (*par*A and *par*B) and recombination functions (*int*, *int*M, and *ccd*B), supporting its localization within a mobile element, such as a conjugative plasmid or a prophage related structure, facilitating horizontal gene transfer.

To further investigate uncharacterized coding sequences annotated as “NA: KEYword”, we performed in silico functional predictions via EggNOG-mapper. Mapping against both the CARD and EggNOG databases identified that the *bla*NDM-1 gene was flanked by genes annotated as members of the phage integrase family (COG0582, PFAM: Phage_integrase, COG category: L), reinforcing the hypothesis of mobilization via prophage-related mechanisms. This genomic arrangement indicates that prophage-derived elements may contribute to the mobilization or stabilization of the carbapenemase gene in either plasmid or chromosomal contexts.

Additionally, a third plasmid, pAC249 (6894 bp; Figure 4), was identified, carrying only the *mph*A gene, which confers resistance to azithromycin. It contained a single integration/excision element and was classified as non-mobilizable.

Overall, the consistent presence of conserved conjugative plasmids encoding a wide array of AMR genes in all outbreak isolates underscores their central role in resistance dissemination. By contrast, non-mobilizable plasmids appeared more variable and carried fewer resistance determinants, suggesting a more limited contribution to horizontal gene transfer.

### 2.6. CRISPR-Cas Loci Characterization

The presence and characteristics of CRISPR-Cas systems were analyzed in six panresistant *K. pneumoniae* ST392 strains. A bioinformatic analysis via CCtyper identified the presence of Type I-E CRISPR-Cas systems in all strains investigated, albeit with significant variations in their composition and potential functionality (Table 3).

Notably, two distinct Type I-E CRISPR-Cas loci were identified based on their consensus repeat sequences: Locus 1 (consensus repeat: GTGTTCCCCGCGCCAGCGGGGATAAACCG) and Locus 2 (Consensus Repeat: CGGTTTATCCCCGCTGGCGCGGGGAACAC). Locus 1 was detected in the strains HJM-EMC466, HJM-NEE391, and HJM-ERG332. All three strains presented a complete and well-conserved set of cas genes characteristic of the Type I-E system (including *Cys*H, *Cas3*, *Cse1*, *Cse2*, *Cas7*, *Cas5*, *Cas6*, *Cas1*, and *Cas2*), with high sequence coverage values and low E-values. These findings suggest that the components of this Type I-E system are genetically intact and potentially functional in these strains. However, variability was observed in the number of repeats and, consequently, spacers. Strain 466 presented 11 repeats (10 spacers), whereas the strains HJM-NEE391 and HJM-ERG332 presented 44 repeats (43 spacers each), indicating a greater expansion of the CRISPR memory region in the latter.

The second Type I-E CRISPR-Cas, Locus 2 was identified in the strains HJM-ICO381, HJM-CBA208, HJM-CCQ423, and as a newly identified orphan locus in strain HJM-EMC466. Strains HJM-CBA208 and HJM-CCQ423 presented a complete and conserved complement of Type I-E *cas* genes, with a repeat identity of 100%, an average of 44 repeats (43 spacers), and an average spacer length of 32 bp. This finding indicates that the Type I-E system associated with this locus is also genetically intact and potentially active in these two strains.

A crucial finding was the characterization of Locus 2 in strain HJM-ICO381, which contained two distinct CRISPR arrays corresponding to this locus. The first array comprised 30 repeats (29 spacers), whereas the second array included 10 repeats (9 spacers). Importantly, no associated *cas* genes were detected for either of these CRISPR loci in the genome of strain HJM-ICO381, indicating that both represent orphan CRISPR arrays. This likely signifies functional inactivity of the adaptive defense mechanism, suggesting that the observed panresistance in strain HJM-ICO381 may be attributed, at least in part, to the absence of a functional CRISPR-Cas system capable of mediating defense against the acquisition of mobile genetic elements.

Similarly, the newly identified orphan Locus 2 in strain HJM-EMC466, with 34 repeats and no associated *cas* genes, further exemplifies this pattern of incomplete CRISPR-Cas systems within the analyzed isolates.

For the strains HJM-EMC466, HJM-NEE391, HJM-ERG332, HJM-CBA208, and HJM-CCQ423, the presence of Type I-E CRISPR-Cas systems with intact *cas* genes, very low E-values, and high coverage indicates that these systems are genetically functional. Despite this apparent adaptive defense capability, all these strains presented a panresistant phenotype, including the confirmed presence of *bla*NDM on a plasmid and other resistance genes on a different plasmid. These findings suggest that the functionality of the CRISPR-Cas system is insufficient to counteract the acquisition and persistence of resistance plasmids in these strains. Possible explanations include the absence of specific spacers targeting the identified resistance plasmids, the acquisition of resistance plasmids before the expansion of the CRISPR memory or during a transient state of system inactivity, or the existence of evasion mechanisms by the plasmids (e.g., anti-CRISPR genes or mutations in proto-spacer or PAM sequences).

Notably, none of the analyzed CRISPR-Cas loci contained spacers targeting sequences from the plasmid carrying *bla*NDM, despite its widespread dissemination among the isolates.

The genomic organization of the CRISPR-Cas type I-E loci, including the arrangement of *cas* genes and CRISPR arrays, is illustrated in Figure 5.

### 2.7. Phylogenetic Analysis

To assess the genetic relatedness of the isolates and to confirm the clonal nature of the outbreak, a phylogenetic analysis was performed using the autoMLST platform based on core genome multilocus sequence typing. This study included a reference dataset of *K. pneumoniae* genomes from diverse geographical and clinical sources. All six outbreak isolates are clustered within a well-supported monophyletic clade, confirming a common origin and suggesting clonal dissemination within the hospital setting.

A comparative genomic analysis identified close phylogenetic proximity to *K. pneumoniae* subsp. *pneumoniae* NTUH-K2044 (GCF_000009885.1), a well-characterized hypervirulent clinical strain, supports their classification of the isolates within the *K. pneumoniae* species complex (KpSC). These findings underscore the clinical importance of the outbreak strain, and raise concerns regarding its potential virulence and transmission dynamics (Figure 6).

### 2.8. Virulence Analysis

Screening with Kleborate and Abricate identified a broad repertoire of virulence-associated genes despite the absence of classical hypervirulence markers (*rmp*A/*rmp*A2, *iuc*A, *iro*B, *clb*, and peg-344). All isolates carried multiple adhesin-encoding genes, including *fim*A and *fim*H (type 1 fimbriae), *mrk*ABCDF (type 3 fimbriae), *ecp*A, and *kpn*, which facilitate adherence to epithelial cells and abiotic surfaces. The mrk operon encodes type 3 fimbriae, filamentous structures crucial for biofilm formation and bacterial adherence to medical devices and host tissues, thereby contributing to persistence and virulence in nosocomial infections. Genes involved in capsule biosynthesis (*wzi*1187 and *wzc*) contribute to immune evasion. Biofilm-associated genes were present in all isolates, including *pga*ABCD and *bcs*A, which are implicated in extracellular polysaccharide production and surface attachment. Iron acquisition systems were abundant: the enterobactin operon (*ent*ABCDEF, *fep*A, *fep*B, and *fep*C) was present in all strains, along with *iut*A (aerobactin receptor) and *kfu*ABC (iron uptake). Genes encoding Type VI secretion system components (*hcp* and *vgr*G) were consistently detected, suggesting potent roles in bacterial competition and host interaction. Outer membrane protein genes (*omp*A, *ycf*M, and *tol*C) associated with serum resistance and efflux were also identified. Overall, the virulence gene profile suggests that these ST392 strains possess a multifactorial pathogenic potential adapted for persistence and dissemination in the hospital environment rather than hypervirulence in community-acquired infections.

To evaluate the hypervirulence potential of the isolates, only the *iut*A gene, encoding the ferric aerobactin receptor, was detected. However, Kleborate assigned a virulence score of 0/5, as none of the hallmark hypervirulence loci-YbST (yersiniabactin), AbST (aerobactin), SmST (Salmochelin), or CbST (colibactin) were identified. For capsular typing, only the *wzi*187 allele was detected, with no association with hypervirulent K-serotypes, such as K1 or K2. Therefore, the isolates do not meet the genomic criteria for classification as hypervirulent *K. pneumoniae* strains.

### 2.9. Global cgMLST Phylogeny

A phylogenetic analysis of 200 *K. pneumoniae* ST 392 genomes was performed via the cgMLST scheme to determine the genetic relationships among the strains and to contextualize the diversity of the Mexican strains. The resulting maximum parsimony tree (Figure 7) identified considerable genetic diversity within the ST392 lineage, with groupings suggesting both global dissemination and regional transmission events.

The reference strains included in the analysis originated from diverse geographical locations, including Spain, the United Kingdom, Nigeria, Portugal, the USA, Germany, Switzerland, Singapore, the Netherlands, Colombia, Norway, Tunisia, Lebanon, Israel, India, and Uruguay. This broad geographical representation allowed for a robust evaluation of global ST392 diversity.

An analysis of the Mexican strains identified that they do not form a discrete, monophyletic clade, indicating multiple introductions or underlying genetic diversity. The strains clustered within a phylogenetic branch that demonstrated a close relationship with strains predominantly from Spain (SAMEA14430890) and with a group of strains from the USA (SAMN 12393011). This co-clustering indicates a possible epidemiological connection or shared ancestry with lineages circulating in both European and North American contexts.

## 3. Discussion

This study reports a hospital outbreak caused by *K. pneumoniae* ST392, a member of clonal complex 147 (CC147), characterized by pandrug-resistance and high mortality. Isolates were obtained from patients with severe infections, including bloodstream infections, ventilator-associated pneumonia, and liver abscesses, many of whom had prior exposure to carbapenems or third-generation cephalosporins. The rapid dissemination of clonally related strains across multiple hospital departments indicates a failure in infection control, supporting the classification of this event as an actual outbreak.

All isolates exhibited resistance to both first- and last-line antimicrobials, including carbapenems and colistin. A genomic analysis identified the presence of *bla*NDM-1, *bla*OXA-1, *bla*TEM-1, and *bla*CTX-M-15, along with 33 additional resistance genes conferring resistance to 10 antibiotic classes. Colistin resistance was associated with a chromosomal *pmr*B R256G mutation, which alters lipid A and reduces colistin binding [16]. However, our study did not include functional validation of the *pmr*B R256G mutation. Therefore, while the literature supports its role [17,18,19,20], its contribution in these isolates remains inferential and warrants experimental confirmation.

A plasmid analysis identified two key conjugative plasmids (pAA046 and pAA276) encoding major resistance determinants, including *bla*NDM-1, and *rmt*C and *aac(6′)-Ib-cr*, respectively. Both plasmids carried multiple mobile genetic elements involved in integration/excision, replication, and horizontal transfer. Interestingly, *bla*NDM-1 was flanked by phage integrase genes (COG0582), suggesting mobilization through prophage-related mechanisms. This is consistent with recent evidence underscoring the role of phage–plasmid hybrids in disseminating clinically relevant resistance genes [21,22]. These elements may enhance the stability and horizontal transfer of resistance genes among diverse bacterial species.

Additionally, we identified a small, non-conjugative plasmid (pAC249) that harbors only *mph*A, which confers resistance to azithromycin. Despite lacking mobilization genes, similar small plasmids have been documented in *Escherichia coli* and *Salmonella enterica*, and are associated with azithromycin resistance despite lacking known mobilization elements [23,24]. These findings suggest that even non-conjugative plasmids may play a role in the persistence and dissemination of macrolide resistance under selective pressure, potentially through co-selection with other resistance determinants or stable maintenance in host bacteria.

An integron analysis identified both CALIN and In0 structures, reflecting a dynamic history of gene cassette acquisition and loss. Although classical class 1 integrons were absent, some CALIN structures demonstrated 100% identity to known plasmids carrying resistance cassettes, such as *aad*A5 and *dfr*A17 [25,26]. These findings support the ongoing microevolution and remodeling of integrons in response to antimicrobial pressure, enhancing the adaptability of *K. pneumoniae* in healthcare environments.

Type I-E CRISPR-Cas systems were detected in all isolates; however, no evidence indicated that these systems restricted the acquisition of *bla*NDM carrying plasmids. The absence of spacers targeting *bla*NDM is consistent with previous studies, which suggest that CRISPR-Cas systems in hospital-adapted *K. pneumoniae* are frequently inactive or evaded by mobile elements [27,28]. A structural analysis demonstrated considerable diversity: while some strains carried complete CRISPR-Cas loci with intact adaptation and interference modules, others had orphan arrays lacking *cas* genes or exhibited fragmented operons. The detection of multiple loci in the strain HJM-EMC466 indicates possible redundancy or partial degradation, a phenomenon observed in other *Enterobacterales* lineages [29].

The persistence of panresistance despite the presence of intact CRISPR-Cas loci indicates functional inactivity or evasion by resistance plasmids, potentially through absence of protospacers, anti-CRISPR mechanisms, or acquisition during CRISPR-inactive periods [30]. A spacer profile analysis supported the notion that clonal expansion was the primary driver of dissemination, consistent with the lineage-specific patterns observed in ST392 and ST147 outbreaks [31].

A phylogenetic analysis confirmed that the six outbreak isolates formed a monophyletic clade, supporting a clonal origin and intra-hospital transmission. The cgMLST comparisons identified that the Mexican strains clustered closely with isolates from Spain and the USA, suggesting shared ancestry or international dissemination. The presence of genetic diversity among the local ST392 strains also implies multiple introductions or microevolutionary divergence within Mexico.

Interestingly, the outbreak isolates were closely related to *K. pneumoniae* subsp. *pneumoniae* NTUH-K2044, a reference hypervirulent strain, although they lacked classical virulence determinants, such as *rmp*A/*rmp*A2, siderophores (aerobactin or salmochelin), and the K1/K2 capsular serotype [32]. This indicates that, although not hypervirulent, these ST392 strains may have acquired genetic features that enhance their fitness in the hospital environment, including adhesins, biofilm formation, or type VI secretion systems [2].

The capsular type KL27 and *wzi*187 allele, found in all isolates, is rare but has been previously reported in ST392 outbreaks and even in zoonotic cases, raising concerns about cross-species transmission or environmental reservoirs [33]. These findings underscore the importance of a One Health approach in understanding the dissemination of resistance.

The widespread colistin resistance observed in this outbreak highlights the need for alternative therapies. Promising strategies include LpxC inhibitors, bacteriophage-based treatments, and novel antibiotics, like cefiderocol, which have shown efficacy against MDR and colistin-resistant Gram-negative pathogens [34,35,36,37].

To our knowledge, this is the first comprehensive report in Mexico documenting an outbreak caused by *K. pneumoniae* ST392 with pandrug-resistance and complete genomic characterization. These findings are essential for understanding the dissemination dynamics of high-risk clones and for guiding local and regional containment strategies.

Although the isolates were obtained in 2016, their genomic features remain relevant due to the ongoing global spread of the *bla*NDM-harboring strains and the continued clinical challenges posed by panresistant *K. pneumoniae*. A key limitation of this study is the retrospective nature of the sampling and the lack of functional assays to confirm the inferred resistance mechanisms, such as the activity of the CRISPER-Cas system and the role of *pmr*B R256G. Nevertheless, the comprehensive genomic dataset provides valuable insights into the architecture and mobility of antimicrobial resistance determinants in ST392. We acknowledge that the presence of CRISPR-Cas systems alone does not imply functional activity. In this study, no experimental assays were performed to confirm CRISPR expression or plasmid interference. Therefore, our interpretation is limited to genomic characterization, and the role of CRISPR-Cas in plasmid acquisition remains hypothetical. Nonetheless, the structural variability observed across the isolates provides a useful genomic context to understand the microevolution of ST392 during the outbreak.

Although the isolates were obtained in 2016, their genomic features remain relevant due to the ongoing global spread of the *bla*NDM-harboring strains. A limitation of this study is the lack of functional assays to validate inferred resistance mechanisms, including CRISPR-Cas activity and the role for *pmr*B R256G. We acknowledge that a temporal phylogenetic analysis to estimate divergence times and the relationship to NTUH-K2044 was not performed, as it was beyond the primary scope of this work. Such analyses will be considered in future studies aimed at reconstructing the evolutionary history of ST392.

The findings are particularly relevant for hospital settings in Latin America, where genomic surveillance is limited and infection control challenges persist. The identification of clonally disseminated, plasmid-driven, pandrug-resistant *K. pneumoniae* strains underscores the urgency of integrating whole-genome sequencing into hospital-based antimicrobial resistance surveillance programs. Strengthening infection control protocols and implementing routine genomic monitoring could play a crucial role in limiting the spread of high-risk clones in tertiary care institutions.

## 4. Materials and Methods

### 4.1. Bacterial Isolation and Antimicrobial Susceptibility Testing

Six clinical isolates of carbapenem-resistant *K. pneumoniae*, obtained from hospitalized patients between January and September 2016, were analyzed. The strain HJM-EMC466 was identified as the index case of the outbreak, while the remaining strains (HJM-NEE391, HJM-ERG332, HJM-ICO381, HJM-CBA208, and HJM-CCQ423) were subsequently recovered in chronological order from clinical samples, including blood, urine, respiratory secretions, and wound swabs. All samples were processed in the hospital’s Bacteriology Laboratory. Patient characteristics are summarized in Table 1.

Bacterial identification was performed via the automated Phoenix-100 (Becton-Dickinson, Franklin Lakes, NJ, USA). Antimicrobial susceptibility testing was conducted using broth microdilution to determine minimum inhibitory concentrations (MICs), and the results were interpreted according to the Clinical and Laboratory Standards Institute (CLSI) guidelines (2021) [38]. Carbapenemase production was phenotypically confirmed using the TEST/CARBA-5 assay (NG BIOTECH, Guipry, France).

### 4.2. Whole-Genome Sequencing and Bioinformatic Analysis

Genomic DNA was extracted using the QIAamp DNA Blood Mini Kit (Qiagen, Hilden, Germany). Library preparation was performed using the Nextera XT DNA Library Preparation Kit (Illumina, San Diego, CA, USA), and sequencing was conducted on the Illumina MiSeq platform with paired-end reads. Quality control of the raw reads was assessed via FastQC, and de novo assembly was performed with SPAdes. Genome annotation was conducted using Prokka (version 1.14.6).

Antimicrobial resistance genes and mobile genetic elements were identified via ResFinder [39], AMRFinderPlus (3.11.20), MOBsuite (3.1.9), mobileOG-db (beatrix-1.6) [40], Kleborate (v1.0.229), and IntegronFinder (via Galaxy-Pasteur) [41]. Plasmid content and mobility were assessed using PROKSEE [42,43]. Virulence genes were detected using Abricate (1.0.1) and EggNOG-MAPPER, a genome-wide functional annotation tool. Capsular (KL) and lipopolysaccharide (O) locus typing were performed using the BIGSdb-Kp platform [44] and Kleborate. CRISPR-Cas loci were identified and characterized via CRISPRCasFinder [45] and CCtyper. Spacer content and locus organization were analyzed to assess the structure and potential functionality of the CRISPR-Cas loci across isolates.

### 4.3. Phylogenetic Analysis

Multilocus sequence typing (MLST) was performed using the MLST tool hosted by the Center for Genomic Epidemiology [46]. Core genome multilocus sequence typing (cgMLST) was performed using the Pathogenwatch platform [47]. Assembled genomes (in FASTA format) were uploaded and analyzed using the ST392-specific reference scheme. The dataset included 6 Mexican isolates and 194 publicly available ST392 genomes from diverse geographic locations, totaling 200 genomes.

Allele calling was performed using a BLASTn-like algorithm with a minimum identity threshold of ≥90% and ≥95% coverage. The cgMLST profiles were used to generate a pairwise distance matrix based on the allele differences. Phylogenetic trees were constructed using maximum parsimony and visualized using MEGA 12 [48] and iTOL [49]. This analysis allowed for high-resolution comparisons of genetic relationships among the local and international ST392 strains.

## 5. Conclusions

These findings confirm the clonal spread of pandrug-resistant *K. pneumoniae* ST392-KL27 in a Mexican hospital, driven by conjugative plasmids and limited CRISPR-Cas activity. The presence of virulence-associated genes and global phylogenetic relatedness highlights the strain’s adaptability and potential for international dissemination, underscoring the need for enhanced surveillance and infection control measures.

## Figures and Tables

**Figure 1 ijms-26-08047-f001:**
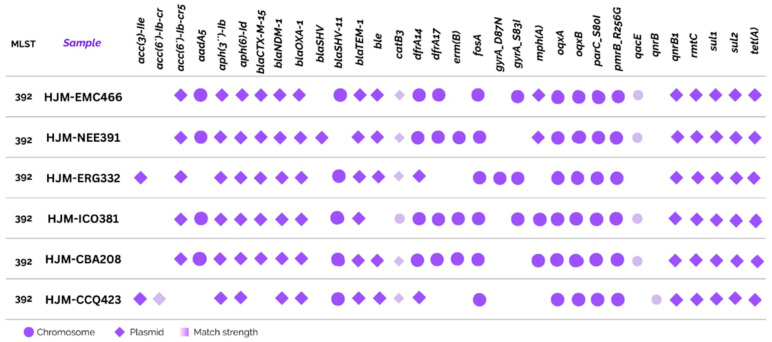
Resistome of *K. pneumoniae* isolates.

**Figure 2 ijms-26-08047-f002:**
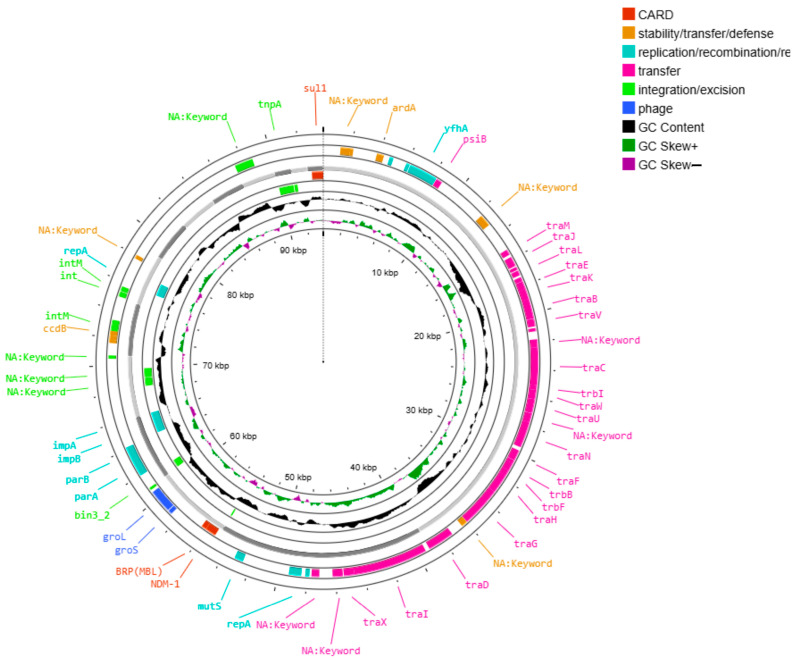
Circular map of the plasmid pAA046. The map illustrates the genetic organization of plasmid AA46 (96,552 bp), which was identified in isolate HJM-EMC466. The outer rings display annotated genes categorized by function, as color-coded in the legend located in the upper right corner of the figure. Several open reading frames (ORFs) surrounding resistance and transfer genes were annotated as “NA: Keyword”, reflecting hypothetical or uncharacterized proteins.

**Figure 3 ijms-26-08047-f003:**
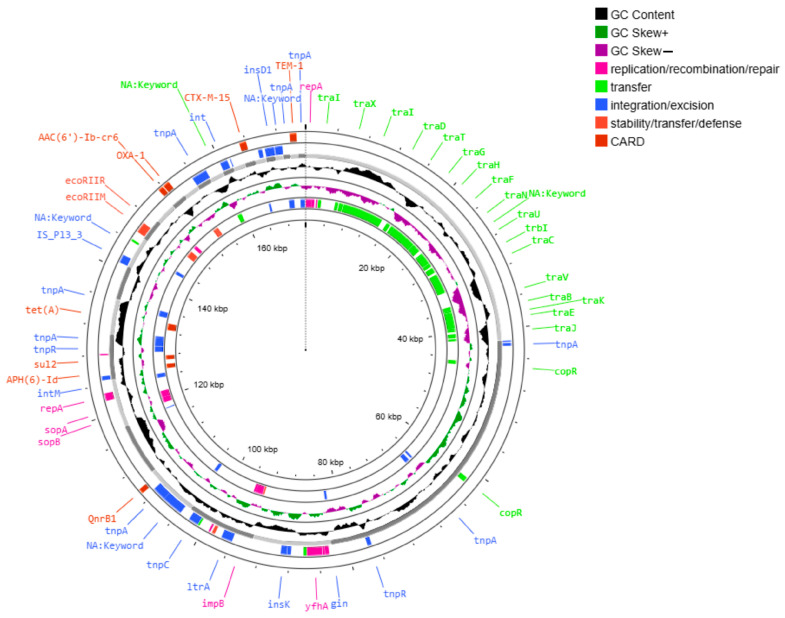
Circular map of the plasmid pAA276. A circular representation of the plasmid pAA276 (172,586 bp), which was identified in multidrug-resistant isolates. The outer rings display functionally annotated genes, with color coding corresponding to the legend shown in the upper right corner of the figure. This plasmid harbors multiple resistance genes, including *aac(6′)-Ib-cr5*, *aph(3″)-Ib*, *aph(6)-Id*, *bla*CTX-M15, *bla*OXA-1, *bla*TEM-1, *qnr*B1, *sul*2, *tet*A, and *cat*B3.

**Figure 4 ijms-26-08047-f004:**
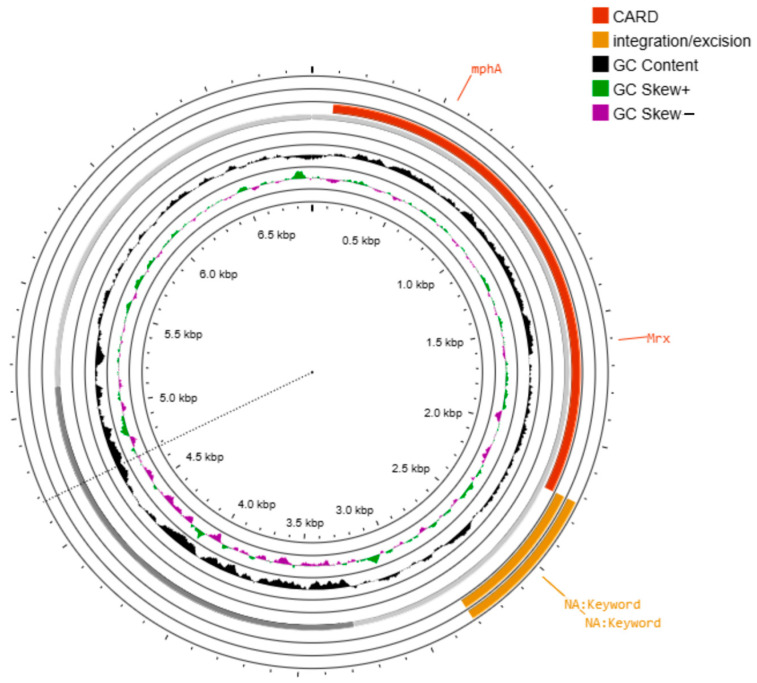
Circular map of the plasmid pAC249. A circular representation of the plasmid pAC249 (6894 bp), showing one mobile genetic element involved in integration/excision, and the resistance gene *mph*A. This plasmid is non-mobilizable.

**Figure 5 ijms-26-08047-f005:**
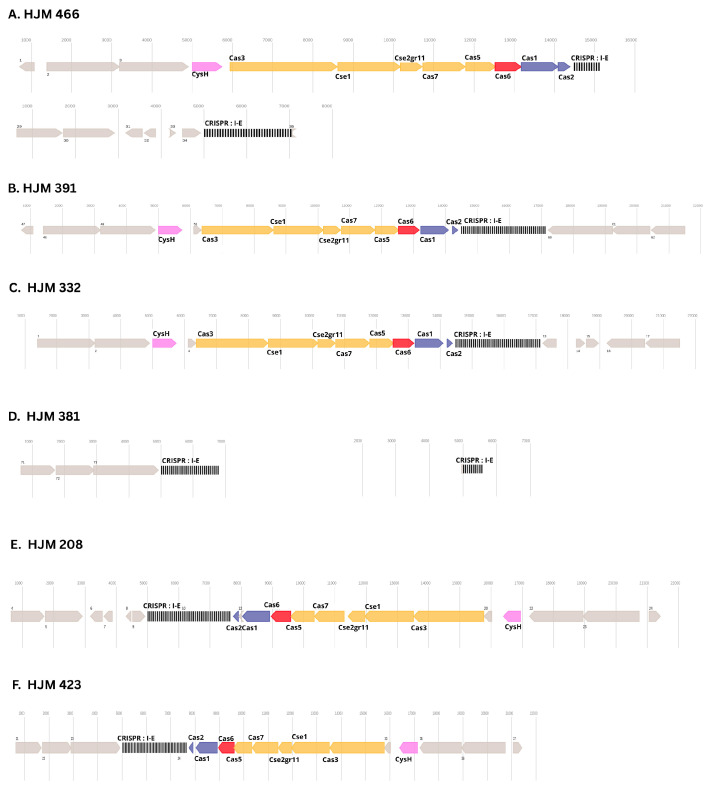
CRISPR-Cas loci, orphan Cas operons, and orphan CRISPR arrays in six ST392 *K. pneumoniae* isolates. Schematic representation of CRISPR-Cas regions in six PDR *K. pneumoniae* ST 392 isolates (**A**–**F**). Interference module genes are shown in yellow, adaptation genes in blue, accessory genes in purple, and Cas6 in red. Arrays are shown in black/white boxes, labeled with their associated subtypes. Genes with unknown functions are shown in gray. Low-confidence Cas gene alignments are displayed in light colors, with parentheses surrounding the gene name. The numbers adjacent to the arrays indicate overlapping (false) ORF-predicted CRISPR arrays. Notably, the strains HJM-391, HJM-332, and HJM-466 display complete Type I-E loci, whereas HJM-381 harbors an orphan CRISPR array, and the strains HJM-208 and HJM-432 show atypical or fragmented CRISPR-Cas loci. These structural differences suggest lineage-specific erosion or divergence of CRISPR-Cas immunity in hospital-adapted *K. pneumoniae*.

**Figure 6 ijms-26-08047-f006:**
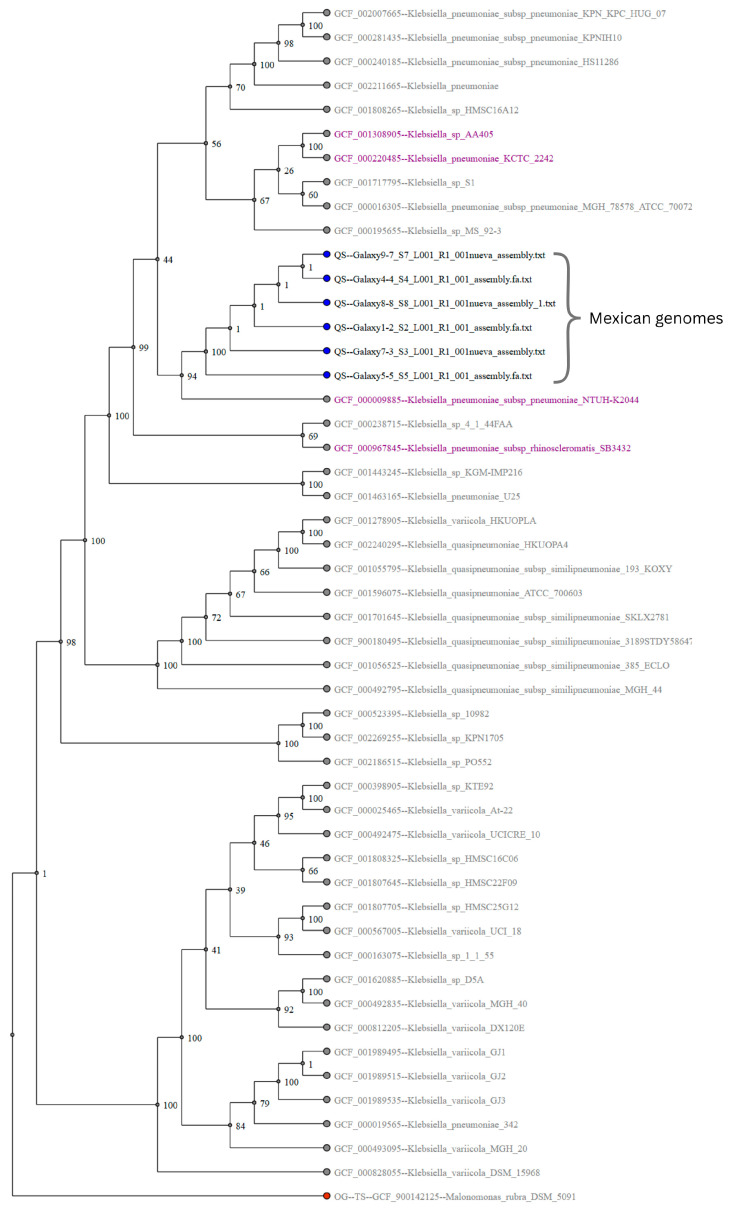
Maximum likelihood phylogenetic tree generated via the autoMLST platform based on core-genome multilocus sequence typing. The six outbreak isolates are clustered within a single monophyletic clade, supporting their clonal origin. Reference genomes were automatically selected by the platform based on phylogenetic proximity. Notably, *K. pneumoniae* subsp. *pneumoniae* NTUH-K2044 (a hypervirulent strain) are located near the outbreak clade, indicating close evolutionary relationships within the *K. pneumoniae* species complex (KpSC). Isolates highlighted in purple indicate the presence of siderophore-related genes, suggesting the potential for increased virulence.

**Figure 7 ijms-26-08047-f007:**
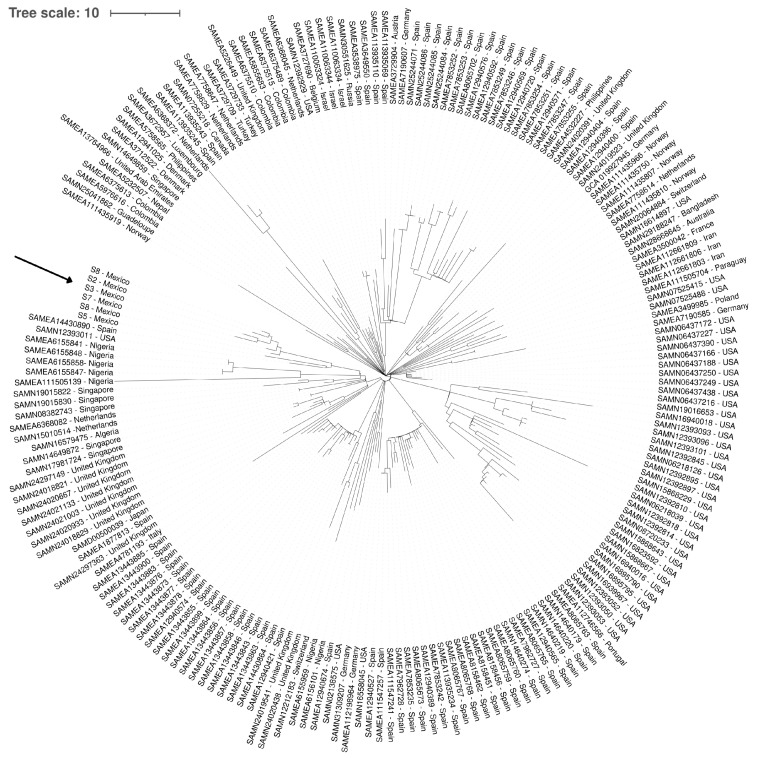
Phylogenetic tree of *K. pneumoniae* ST 392 based on cgMLST analysis of 200 genomes. The tree illustrates the genetic relationships among the strains, including the Mexican strains (S2, S3, S4, S5, S7, and S8), which are highlighted in the text. The branches represent genetic divergence, and the labels at the tips of the branches indicate the strain name and its geographical origin.

**Table 1 ijms-26-08047-t001:** Characteristics of patients from whom *K. pneumoniae* isolates were collected.

	*Patient Gender*	*Age*	*Isolation Source*	*Patient Survival*	*Diagnostic*	*Treatment Before Detection*	*Treatment After Detection*
*HJM-EMC466*	Female	81	Expectoration	No	Liver abscess and pneumoniae	Levofloxacin, metronidazole, cefepime, and ciprofloxacin	Colistin
*HJM-NEE391*	Female	45	Catheter	Yes	Subarachnoid hemorrhage	Ceftriaxone, piperacillin tazobactam and vancomycin	Colistin
*HJM-ERG332*	Male	28	Urine	No	Testicular cancer and pulmonary metastasis	Ciprofloxacin, ceftazidime, and amikacin	Colistin
*HJM-ICO381*	Male	34	Purulent wound	No	Gastric G1 neuroendocrine tumor and septic shock	Piperacillin tazobactam, vancomycin, and imipenem	Colistin
*HJM-CBA208*		32	Blood	No	Soft tissue sepsis, pneumonia, and urinary tract infection	Levofloxacin, metronidazole, meropenem, ceftazidime, and cefepime	Colistin
*HJM-CCQ423*	Male	53	Expectoration	No	Soft tissue infection and uncontrolled *diabetes mellitus*	Imipenem, clindamycin, levofloxacin, ceftriaxone, and metronidazole	Colistin

**Table 2 ijms-26-08047-t002:** Plasmid characterization in *K. pneumoniae* ST392 clinical isolates based on mobility element analysis.

*Isolate*	*Mob* *Cluster*	*Size (BP)*	*Amr Gene*	*Replicon Types*	*Relaxase Types*	*Mash* *Nearest Neighbor*	*Mash Neighbor Distance*	*Mobility*
*HJM-EMC466*	AA276	1,171,674	10	IncFIB, IncFII	MOBF	CP010390	0.00243596	Conjugative
	AA046	93,728	4	IncFIB, IncFII,	MOBF	CP022350	0.00227104	Conjugative
	AA103	4171	-	ColRNAI	-	CP024435	0.000215742	Non-mobilizable
	AC249	6894	1	-	-	KU254581	0.0588483	Non-mobilizable
*HJM-NEE391*	AA276	172,586	10	IncFIB, IncFII	MOBF	CP010390	0.00240836	Conjugative
	AA046	96,552	5	IncFIB, IncFII,	MOBF	CP022350	0.00360762	Conjugative
	AA103	4167	-	ColRNAI	-	CP024435	0.000312573	Non-mobilizable
	AB749	-	-	-	-	LR134256	0.0350415	Non-mobilizable
	ADO94	12,963	1	-	-	CP015133	0.0437856	Non-mobilizable
*HJM-ERG332*	AA276	174,011	12	IncFIB, IncFII	MOBF	CP010390	0.0025468	Conjugative
	AA046	92,509	4	IncFIB, IncFII,	MOBF	CP022350	0.00366686	Conjugative
	AA103	4168	-	ColRNAI	-	CP024435	0.000264084	Non-mobilizable
*HJM-ICO381*	AA276	167,036	9	IncFIB, IncFII	MOBF	CP010390	0.00340187	Conjugative
	AA046	87,014	4	IncFIB, IncFII,	MOBF	CP022350	0.00448671	Conjugative
	AA103	4167	-	ColRNAI	-	CP024435	0.000239895	Non-mobilizable
*HJM-CBA208*	AA276	170,592	10	IncFIB, IncFII	MOBF	CP010390	0.00268639	Conjugative
	AA046	93,856	4	IncFIB, IncFII,	MOBF	CP022350	0.00249129	Conjugative
	AA103	4167	-	ColRNAI		CP024435	0.000483446	Non-mobilizable
*HJM-CCQ423*	AA046	95,436	4	IncFIB, IncFII,	MOBF	CP022350	0.00369655	conjugative
	AA276	172,300	11	IncFIB, IncFII	MOBF	CP010390	0.00218918	conjugative
	AB749	14,407	-	-	-	LR134256	0.0350415	non-mobilizable
	AA103	4170	-	ColRNAI	-	CP024435	0.000191626	non-mobilizable

**Table 3 ijms-26-08047-t003:** Characteristics of Type I-E CRISPR-Cas loci in panresistant *K. pneumoniae* ST392 strains.

*Strain*	CRISPR-Cas Subtype	Consensus Repeat (5′-3′)	N Repeats	Repeat Length (bp)	Avg. Spacer Length (bp)	Repeat Identity (%)	Spacer Identity (%)	Spacer Length SEM	Cas Genes	Trusted
*466*	I-E	GTGTTCCCCGCGCCAGCGGGGATAAACCG	11	29	32	100	42.4	0.1	Yes (complete)	TRUE
	I-E	CGGTTTATCCCCGCTGGCGCGGGGAACAC	34	29	31.9	93.1	39.5	0.1	No (Orphan)	TRUE
*391*	I-E	GTGTTCCCCGCGCCAGCGGGGATAAACCG	44	29	32	98.7	40.9	0.7	Yes (complete)	TRUE
*219*	I-E	GTGTTCCCCGCGCCAGCGGGGATAAACCG	44	29	32	98.7	40.9	0.1	Yes (complete)	TRUE
*381*	I-E	CGGTTTATCCCCGCTGGCGCGGGGAACAC	30	29	31.9	93.1	42.7	0.1	No (Orphan)	TRUE
	I-E	CGGTTTATCCCCGCTGGCGCGGGGAACAC	10	29	32	100	40.5	0.0	No (Orphan	TRUE
*208*	I-E	CGGTTTATCCCCGCTGGCGCGGGGAACAC	44	29	32	100	41.8	0.1	Yes (complete)	TRUE
*423*	I-E	CGGTTTATCCCCGCTGGCGCGGGGAACAC	44	29	32	100	41.8	0.1	Yes (complete)	TRUE

## Data Availability

The whole genome sequences of the *K. pneumoniae* isolates have been deposited in NCBI.BioProject:PRJNA764253. Available online: https://www.ncbi.nlm.nih.gov/bioproject/764253 (accessed on 7 October 2021).

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
