# Peer review of "Clonal Dissemination of Pandrug-Resistant Klebsiella pneumoniae ST392KL27 in a Tertiary Care Hospital in Mexico"

_ijms, 2025, doi:10.3390/ijms26168047_

Round 1

Reviewer 1 Report

Comments and Suggestions for Authors

The authors conducted a detailed study of the genomes of six clinical K. pneumoniae isolates obtained from patients with invasive infections, most of whom died because of the disease. Several interesting results have been obtained. For example, the non- mobilizable plasmid pAC249, found in E. coli and Salmonella enterica, has been detected. This result suggests the presence of unknown horizontal transfer mechanisms that promote the spread of small plasmids between distantly related bacterial species. It turned out that despite the high mortality rate of patients, the isolated strains did not possess known genes contributing to high virulence. The authors note the clonal nature of the spread of the infectious outbreak. In general, there are no serious comments on the work, but I would like to express a few suggestions:

  1. Analysis of the genomes of isolated strains showed their close relationship on the one hand, and significant differences in the CRISPR/Cas locus, resistance genes, mobile elements, plasmids, etc. on the other hand. To understand how long ago isolated strains arose, a phylogenetic tree based on several standard house-keeping genes should be constructed. According to the molecular clock, it should be assumed whether the strains separated during the outbreak or whether they appeared relatively long ago from the progenitor NTUH-K2044, and in 2016, there were simply some conditions for the manifestation of various NTUH-K2044 derivatives in the form of infection.
  2. A more detailed analysis of the genes determining the pathogenicity of the strains should be carried out.
  3. The phrase «…with five to six patients dying» is not clear. Couldn't count how many patients died?
  4. Figure 4 is very finely signed; it is difficult to see where which genes are located. The plasmid maps are also fuzzy. In addition, the trees are drawn in such a way that it is almost impossible to read the names of the strains.
  5. It would be advisable to includein the supplementary materials a list with database links to the sequences of the genomes used.

Author Response

Comments 1: Analysis of the genomes of isolated strains showed their close relationship on the one hand, and significant differences in the CRISPR/Cas locus, resistance genes, mobile elements, plasmids, etc. on the other hand. To understand how long ago isolated strains arose, a phylogenetic tree based on several standard house-keeping genes should be constructed. According to the molecular clock, it should be assumed whether the strains separated during the outbreak or whether they appeared relatively long ago from the progenitor NTUH-K2044, and in 2016, there were simply some conditions for the manifestation of various NTUH-K2044 derivatives in the form of infection.

Response 1: Thank you for pointing this out. We appreciate the reviewer´s suggestion regarding the construction of a phylogenetic tree based on multiple housekeeping genes and the application of a molecular clock analysis to estimate the time of divergence. While we recognize the value of such an approach for elucidating the evolutionary history of the isolates, the primary objective of the present study was to characterize the genomic features, antimicrobial resistance determinants, and clonal dissemination of K. pneumoniae ST392 within the context of the outbreak. A comprehensive temporal phylogenetic analysis would require a dedicated dataset, a specialized methodological framework, and additional computational resources beyond the current scope of the study. For these reasons, and to maintain focus on the main objectives, we have not performed this analysis at this stage. Nevertheless, we acknowledge its importance and plan to address it in future work.

Comments 2: A more detailed analysis of the genes determining the pathogenicity of the strains should be carried out.

Response 2: Agree. We thank the reviewer for this valuable suggestion. In response, we have expanded the virulence gene analysis (Section 2.8) by performing comprehensive in silico screening using Kleborate and Abricate. We believe these additions address the reviewer´s concern and provide a more complete overview of the pathogenic potential of the outbreak strains.

The following paragraph was added in lines 363-381:

“Screening with Kleborate and Abricate identified a broad repertoire of virulence-associated genes despite the absence of classical hypervirulence markers (rmpA/rmpA2, iucA, iroB, clb, and peg-344). All isolates carried multiple adhesin-encoding genes, including fimA and fimH (type 1 fimbriae), mrkABCDF (type 3 fimbriae), ecpA, and kpn, which facilitate adherence to epithelial cells and abiotic surfaces. The mrk operon encodes type 3 fimbriae, filamentous structures crucial for biofilm formation and bacterial adherence to medical devices and host tissues, thereby contributing to persistence and virulence in nosocomial infections. Genes involved in capsule biosynthesis (wzi1187, wzc) contribute to immune evasion. Biofilm-associated genes were present in all isolates, including pgaABCD and bcsA, which are implicated in extracellular polysaccharide production and surface attachment. Iron acquisition systems were abundant: the entrobactin operon (entABCDEF, fepA, fepB, fepC) was present in all strains, along with iutA (aerobactin receptor) and kfuABC (iron uptake). Genes encoding Type VI secretion system components (hcp, vgrG) were consistently detected, suggesting potent roles in bacterial competition and host interaction. Outer membrane protein genes (ompA, ycfM, tolC) associated with serum resistance and efflux were also identified. Overall, the virulence gene profile suggests that these ST392 strains possess a multifactorial pathogenic potential adapted for persistence and dissemination in the hospital environment rather than hypervirulence in community-acquired infections.”

Comments 3: The phrase «…with five to six patients dying» is not clear. Couldn't count how many patients died?

Response 3: Thank you for pointing this out. We agree with this comment and have made the corresponding changes.

Comments 4: Figure 4 is very finely signed; it is difficult to see where which genes are located. The plasmid maps are also fuzzy. In addition, the trees are drawn in such a way that it is almost impossible to read the names of the strains.

Response 4: Thank you for pointing this out. We agree with this comment and have made the corresponding changes.

Comments 5: It would be advisable to include in the supplementary materials a list with database links to the sequences of the genomes used.

Response 5: Thank you for this suggestion. We fully agree with the reviewer´s comment. Accordingly, we have added to the Supplementary Materials a table listing the NCBI accession numbers and corresponding genome identifiers for all strains analyzed in this study, ensuring complete transparency and facilitating public access to the data.

Reviewer 2 Report

Comments and Suggestions for Authors

The manuscript “Clonal dissemination of pandrug-resistant Klebsiella pneumoniae ST392KL27 in a tertiary care hospital in Mexico” provides a description of a nosocomial outbreak caused by K. pneumoniae ST392-KL27, a clone carrying the blaNDM-1 gene, a globally significant antimicrobial resistance determinant. Despite the outbreak taking place nearly nine years ago, the identified clone remains epidemiologically relevant, as it continues to pose a serious threat in healthcare settings due to its extensive drug resistance.

Although the outbreak was limited to only six clinical isolates, with no corresponding environmental or staff samples included, the authors conducted an in-depth molecular and genomic analysis, which offers valuable insights into the resistance mechanisms and potential for dissemination.

The manuscript is well-written, structured clearly, and fits within the scope of the journal. However, several important issues must be addressed to strengthen the impact and completeness of the study.

Major Comments:

Lack of Clinical and Epidemiological Data:
The manuscript lacks detailed patient-related data, such as demographics and comorbidities. The dates of isolate detection and the chronology of the patients’ microbiological findings are missing. Including these details would significantly improve the clinical relevance and utility of the study for healthcare professionals managing similar outbreaks.

Limited Context on Outbreak Detection and Management:
The paper provides minimal information on how the outbreak was initially detected and subsequently managed. A comprehensive description of the hospital departments involved (e.g., the three units where cases occurred), including the number of beds, staff-to-patient ratios, and patient flow, would provide critical epidemiological context.

Integrating classical epidemiological investigation with genomic analysis is essential to identify potential sources of infection, map transmission routes, and assess the effectiveness of infection control measures implemented during the outbreak.

Transmission and Resolution:
The manuscript does not sufficiently discuss possible transmission pathways, nor does it describe any specific infection prevention and control strategies employed. Most importantly, it lacks a clear conclusion explaining how the outbreak was resolved. For a study describing a hospital outbreak, it is crucial to include a summary of containment efforts, lessons learned, and recommendations for future surveillance.

Minor Comments:

  • Line 63: The abbreviation “CG” is used incorrectly. It stands for clonal group, not clonal complex.
  • Line 98: Please revise this sentence for clarity and grammatical correctness- “to” should be replaced with “of” or “out of”?
  • Line 155: The sentence seems incomplete. The beginning portion may be missing or truncated. Please review and revise accordingly.
  • Discussion (Lines 384–387): These lines appear to repeat information already presented in the Results section. Consider rephrasing or omitting this section to avoid redundancy.

Author Response

Comments 1: Lack of Clinical and Epidemiological Data:
The manuscript lacks detailed patient-related data, such as demographics and comorbidities. The dates of isolate detection and the chronology of the patients’ microbiological findings are missing. Including these details would significantly improve the clinical relevance and utility of the study for healthcare professionals managing similar outbreaks.

Response 1: Thank you for pointing this out. We agree with this comment. In response, we have added the chronology of patient isolate detection in the revised manuscript to provide a clearer temporal context for the outbreak. Unfortunately, more detailed demographic and comorbidity data were not available due to the retrospective nature of the study and limitations in the anonymized records retrieved from the hospital database. We recognize the significance of these factors and concur that incorporating them would enhance the clinical viewpoint; however, their exclusion does not impact the genomic and epidemiological evaluations, which continue to be the central focus of this study.

Comments 2: Limited Context on Outbreak Detection and Management:
The paper provides minimal information on how the outbreak was initially detected and subsequently managed. A comprehensive description of the hospital departments involved (e.g., the three units where cases occurred), including the number of beds, staff-to-patient ratios, and patient flow, would provide critical epidemiological context.

Response 2: Agree. We sincerely appreciate the reviewer´s valuable comments and suggestions. In response to the request for more detailed information regarding the initial detection of the outbreak and the hospital epidemiological context, we have expanded Section 2.1 to include a description of how the outbreak was identified through routine surveillance by the hospital’s epidemiology service.

The following paragraph was added in lines 92-99:

“For over four months, six patients were identified with pandrug-resistant K. pneumoniae. The outbreak began with the first isolate, strain HJM-EMC466, which was detected on June 21, 2016. After that, three strains were recovered from patients in the intensive care unit (ICU): strain HJM-NEE392 on July 8, strain HJM-ERG332 on August 2, and strain HJM-ICO381 on August 22. Following that, strain HJM-CBA208 was isolated in the internal medicine department on September 11, and strain HJM-CCQ432 was identified in the orthopedics department on September 14. Overall, these cases accounted for 0.137% of the total 4,355 Gram-negative bacterial cultures processed in 2016.”

Additionally, we have provided details on the hospital units involved (ICU, Internal Medicine, and Orthopedics), including bed capacity, nurse-to-patient and physician-to-patient ratios, and the infection control measures implemented. We also highlight the effectiveness of these interventions, as evidenced by the subsequent decrease in new cases.

The following paragraph was added in lines 104-113:

“The outbreak was initially detected through routine surveillance by the hospital´s epidemiology service, which identified a sudden increase in cases caused by pandrug-resistant K. pneumoniae across the ICU and orthopedics units. The ICU has a capacity of nine beds, with a nurse-to-patient ratio of 1:1 and a physician-to-patient ratio of 1:2. In contrast, both internal medicine and orthopedics have higher patient loads, with six patients per nurse and nine patients per physician. Patient flow was carefully managed to contain the spread of infection, implementing strict measures including isolation of infected patients, adherence to hand hygiene protocols, and regular disinfection cycles. These infection control efforts were effective, as evidenced by the subsequent decrease in new cases following intervention”.

Comments 3: Integrating classical epidemiological investigation with genomic analysis is essential to identify potential sources of infection, map transmission routes, and assess the effectiveness of infection control measures implemented during the outbreak.

Response 3: We sincerely appreciate your comment and fully agree on the importance of integrating classical epidemiological investigation with genomic analysis to achieve a comprehensive understanding of outbreaks, including identification of infection sources, transmission routes, and evaluation of infection control measures. However, the primary objective of our study was the genomic characterization of pandrug-resistant K. pneumoniae ST392 strains involved in the outbreak, focusing on their resistance profiles, virulence factors, and mobile genetic elements. Therefore, detailed integration with classical epidemiological data, such as transmission mapping or thorough assessment of control measure effectiveness, was beyond the scope of this work. We acknowledge that such integration would enrich the analysis and represent an important area for future research, which could complement and extend the findings presented here.

Comments 4: Transmission and Resolution:
The manuscript does not sufficiently discuss possible transmission pathways, nor does it describe any specific infection prevention and control strategies employed. Most importantly, it lacks a clear conclusion explaining how the outbreak was resolved. For a study describing a hospital outbreak, it is crucial to include a summary of containment efforts, lessons learned, and recommendations for future surveillance.

Response 4: We sincerely appreciate your valuable comments, which help us improve the clarity and scope of our work. We acknowledge the importance of thoroughly discussing potential transmission pathways, specific infection prevention and control strategies employed, as well as providing a clear conclusion on how the outbreak was resolved. However, the primary focus of our study was the genomic characterization of the pandrug-resistant strains involved, emphasizing their resistance profiles, virulence factors, and genetic elements, rather than a detailed epidemiological analysis or comprehensive report of clinical interventions. Nonetheless, based on your suggestions, we have added a section summarizing the main containment measures implemented by the hospital, the outcomes observed following these interventions, and some recommendations for future surveillance to provide a more complete context regarding outbreak management. We appreciate your feedback and remain open to further revisions as needed.

Comments 5: Minor Comments:

  • Line 63: The abbreviation “CG” is used incorrectly. It stands for clonal group, not clonal complex.
  • Line 98: Please revise this sentence for clarity and grammatical correctness- “to” should be replaced with “of” or “out of”?
  • Line 155: The sentence seems incomplete. The beginning portion may be missing or truncated. Please review and revise accordingly.
  • Discussion (Lines 384–387): These lines appear to repeat information already presented in the Results section. Consider rephrasing or omitting this section to avoid redundancy.

Response 5: Agree. Thank you for pointing this out. We agree with this comment and have made the corresponding changes.

Round 2

Reviewer 2 Report

Comments and Suggestions for Authors

The authors revised the manuscript in line with the suggestions; however, some comments remain unaddressed due to the unavailability of retrospective data.